# Intraclass clustering: an implicit learning ability that regularizes DNNs

**Simon Carbonnelle, Christophe De Vleeschouwer**
FNRS research fellows
ICTEAM, Universit catholique de Louvain
Louvain-La-Neuve, Belgium
simon.carbonnelle@gmail.com, christophe.devleeschouwer@uclouvain.be

## Abstract

Several works have shown that the regularization mechanisms underlying deep neural networks' generalization performances are still poorly understood (Neyshabur et al., 2015; Zhang et al., 2017). In this paper, we hypothesize that deep neural networks are regularized through their ability to extract meaningful clusters among the samples of a class. This constitutes an implicit form of regularization, as no explicit training mechanisms or supervision target such behaviour. To support our hypothesis, we design four different measures of intraclass clustering, based on the neuron- and layer-level representations of the training data. We then show that these measures constitute accurate predictors of generalization performance across variations of a large set of hyperparameters (learning rate, batch size, optimizer, weight decay, dropout rate, data augmentation, network depth and width).

## 1 Introduction

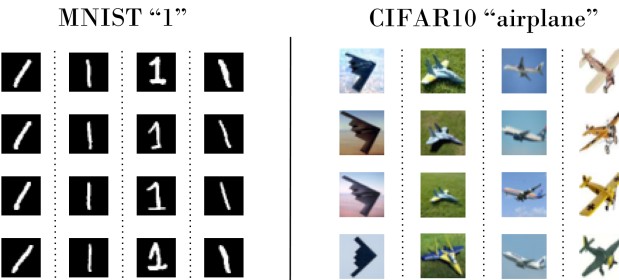

MNIST "1"      CIFAR10 "airplane"

Figure 1: In standard image classification datasets, classes are typically composed of multiple clusters of similarly looking images. We call intraclass clustering a model's ability to differentiate such clusters despite their association to identical labels.

The generalization ability of deep neural networks remains largely unexplained. In particular, the traditional view that explicit forms of regularization (e.g. dropout, $\mathcal{L}_2$-regularization, data augmentation) are the sole factors for generalization performance of state of the art neural networks has been experimentally invalidated (Neyshabur et al., 2015; Zhang et al., 2017). Today's conventional wisdom rather conjectures the presence of implicit forms of regularization, emerging from the interactions between neural network architectures, optimization, and the inherent structure of the data itself (Arpit et al., 2017).

One structural component that seems to occur in most image classification datasets is the presence of multiple clusters amongst the samples of a class (or *intraclass clusters*, cfr. Figure 1). The extraction of such structure in the context of supervised learning is not self-evident, as today's standard training algorithms are designed to group samples from a class together, without any considerations for eventual intraclass clusters.

*This paper hypothesizes that the identification of intraclass clusters emerges during supervised training of deep neural networks, despite the absence of supervision or explicit training mechanisms targeting this behaviour. Moreover, our study suggests that this phenomenon improves the generalization ability of deep neural networks, hence constituting an implicit form of regularization.*

To verify our hypotheses, we define four measures of intraclass clustering and inspect the correlation between those measures and a network's generalization performance. These measures are designed to capture intraclass clustering from four different perspectives, defined by the representation level (neuron vs. layer) and the amount of knowledge about the data's inherent structure (datasets with or without hierarchical labels). To evaluate these measures' predictive power, we train more than 500 models, varying standard hyperparameters in a principled way in order to generate a wide range of generalization performances. The measures are then evaluated qualitatively through visual inspection of their relationship with generalization and quantitatively through the granulated Kendall rank-correlation coefficient introduced by Jiang et al. (2020). Both evaluations reveal a tight connection between intraclass clustering measures and generalization ability, providing important evidence to support this work's hypotheses.

## 2  MEASURING INTRACLASS CLUSTERING IN INTERNAL REPRESENTATIONS

A challenge of our work resides in measuring a model's ability to differentiate intraclass clusters, without knowing which mechanisms underlie such ability. In such context, the design of multiple complementary measures (i) offers different perspectives that will help better characterize intraclass clustering mechanisms and (ii) reduces the risk that their potential correlation with generalization is induced by other phenomena independent of intraclass clustering. This section thus describes four measures of intraclass clustering, differing in terms of representation level (neuron vs. layer) and the amount of knowledge about the data's inherent structure (datasets with or without hierarchical labels).

### 2.1  TERMINOLOGY AND NOTATIONS

$D$ denotes the training dataset. Let $I$ be the number of classes in the dataset $D$, we denote the set of samples from class $i$ by $C_i$ with $i \in \mathcal{I} = \{1, 2, ..., I\}$. In the case of hierarchical labels, $C_i$ denotes the samples from subclass $i$ and $S_{s(i)}$ the samples from the superclass containing subclass $i$. We denote by $\mathcal{N} = \{1, 2, ..., N\}$ and $\mathcal{L} = \{1, 2, ..., L\}$ the indexes of the $N$ neurons and $L$ layers of a network respectively. Neurons are considered across all the layers of a network, not a specific layer. The methodology by which indexes are assigned to neurons or layers does not matter. We further denote by $\text{mean}_{j \in \mathcal{J}}$ and $\text{median}_{j \in \mathcal{J}}$ the mean and median operations over the index $j$ respectively. Moreover, $\text{mean}_{j \in \mathcal{J}}^{k}$ corresponds to the mean of the top-$k$ highest values, over the index $j$.

We call pre-activations (and activations) the values preceding (respectively following) the application of the ReLU activation function (Nair & Hinton, 2010). In our experiments, batch normalization (Ioffe & Szegedy, 2015) is applied before the ReLU, and pre-activation values are collected after batch normalization. In convolutional layers, a neuron refers to an entire feature map. The spatial dimensions of such a neuron's (pre-)activations are reduced through a global max pooling operation before applying our measures.

### 2.2  MEASURES BASED ON LABEL HIERARCHIES

The first two measures take advantage of datasets that include a hierarchy of labels. For example, CIFAR100 is organized into 20 superclasses (e.g. flowers) each comprising 5 subclasses (e.g. orchids, poppies, roses, sunflowers, tulips). We hypothesize that these hierarchical labels reflect an inherent structure of the data. In particular, we expect the subclasses to approximately correspond to different clusters amongst the samples of a superclass. Hence, measuring the extent by which a network differentiates subclasses when being trained on superclasses should reflect its ability to extract intraclass clusters during training.

### 2.2.1 Neuron-level subclass selectivity

The first measure quantifies how selective individual neurons are for a given subclass $C_i$ with respect to the other samples of the associated superclass $S_{s(i)}$. Here, strong selectivity means that the subclass $C_i$ can be reliably discriminated from the other samples of $S_{s(i)}$ based on the neuron's pre-activations[1]. Let $\mu_{n,E}$ and $\sigma_{n,E}$ be the mean and standard deviation of a neuron $n$'s pre-activation values taken over the samples of set $E$. The measure is defined as follows:

$$c_1 = \text{median}_{i \in \mathcal{I}} \ \text{mean}_{n \in \mathcal{N}}^k \ \frac{\mu_{n,C_i} - \mu_{n,S_{s(i)} \setminus C_i}}{\sigma_{n,C_i} + \sigma_{n,S_{s(i)} \setminus C_i}} \tag{1}$$

Since we cannot expect all neurons of a network to be selective for a given subclass, we only consider the top-$k$ most selective neurons. The measure thus relies on $k$ neurons to capture the overall network's ability to differentiate each subclass. We noticed slightly better results when using the median operation instead of the mean to aggregate over the subclasses. We suspect this arises from the outlier behaviour of certain subclasses that we observe in Section 4.5.

### 2.2.2 Layer-level Silhouette score

The second measure quantifies to what extent the samples of a subclass are close together relative to the other samples from the associated superclass *in the space induced by a layer's activations*. In other words, we measure to what degree different subclasses can be associated to different clusters in the intermediate representations of a network. We quantify this by computing the pairwise cosine distances[2] on the samples of a superclass and applying the Silhouette score (Kaufman & Rousseeuw, 2009) to assess the clustered structure of the subclasses. Let $silhouette(a_l, S_{s(i)}, C_i)$ be the mean silhouette score of subclass $C_i$ based on the activations $a_l$ of superclass $S_{s(i)}$ in layer $l$, the measure is then defined as:

$$c_2 = \text{median}_{i \in \mathcal{I}} \ \text{mean}_{l \in \mathcal{L}}^k \ silhouette(a_l, S_{s(i)}, C_i) \tag{2}$$

### 2.3 Measures based on variance

To establish the generality of our results, we also design two measures that can be applied in absence of hierarchical labels. We hypothesize that the discrimination of intraclass clusters should be reflected by a high variance in the representations associated to a class. If all the samples of a class are mapped to close-by points in the neuron- or layer-level representations, it is likely that the neuron/layer did not identify intraclass clusters.

### 2.3.1 Variance in the neuron-level representations of the data

The first variance measure is based on standard deviations of a neuron's pre-activations. If the standard deviation computed over the samples of a class is high compared to the standard deviation computed over the entire dataset, we infer that the neuron has learned features that differentiate samples belonging to this class. The measure is defined as:

$$c_3 = \text{mean}_{i \in \mathcal{I}} \ \text{mean}_{n \in \mathcal{N}}^k \ \frac{\sigma_{n,C_i}}{\sigma_{n,D}} \tag{3}$$

A visual representation of the measure is provided in Figure 2.

### 2.3.2 Variance in the layer-level representations of the data

The fourth measure transfers the neuron-level variance approach to layers by computing the standard deviations over the pairwise cosine distances calculated in the space induced by the layer's activations. Let $\Sigma_{l,E}$ be the standard deviation of the pairwise cosine distances between the samples of set $E$ in the space induced by layer $l$. The measure is defined as:

$$c_4 = \text{mean}_{i \in \mathcal{I}} \ \text{mean}_{l \in \mathcal{L}}^k \ \frac{\Sigma_{l,C_i}}{\Sigma_{l,D}} \tag{4}$$

---

[1] In other words, we are interested in evaluating whether the linear projection implemented by the neuron has been effective in isolating a given subclass.

[2] Using cosine distances provided slightly better results than euclidean distances.

To compute this measure, we found it helpful to standardize the representations of different neurons. More precisely, we normalize each neuron's pre-activations to have zero mean and unit variance, then apply a bias and ReLU activation function such that $25\%$ of the samples are activated[3]. This makes the measure invariant to rescaling and translation of each neuron's preactivations.

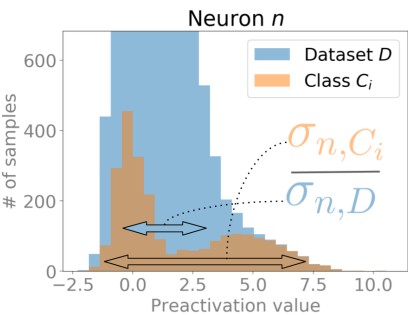

Figure 2: Our simplest measure (denoted $c_3$) quantifies intraclass clustering through the ratio of standard deviations $\sigma_{n,C_i}$ and $\sigma_{n,D}$ associated to the class $C_i$ and the entire dataset $D$ respectively. Intuitively, a high ratio means that the neuron relies on features that *differentiate* samples from $C_i$ although they belong to the same class. Despite its simplicity, our results in Section 4 suggest a remarkably strong connection between $c_3$ and generalization performance. This illustration of measure $c_3$ is based on a neuron from our experimental study, and the associated ratio is $2.47$.

## 3 EXPERIMENTAL SETUP

The purpose of our experimental endeavour is to assess the predictive power of the proposed intraclass clustering measures with respect to the generalization performance. To this end, we reproduce the methodology introduced by Jiang et al. (2020). First of all, this methodology puts emphasis on the scale of the experiments to improve the generality of the observations. Second, it tries to go beyond standard measures of correlation, and puts extra care to detect causal relationships between the measures and generalization performance. This is achieved through a systematic variation of multiple hyperparameters when building the set of models to be studied, combined with the application of principled correlation measures.

### 3.1 BUILDING A SET OF MODELS WITH VARYING HYPERPARAMETERS

Our experiments are conducted on three datasets and two network architectures. The datasets are CIFAR10, CIFAR100 and the coarse version of CIFAR100 with 20 superclasses (Krizhevsky & Hinton, 2009). The two network architectures are Wide ResNets (He et al., 2016; Zagoruyko & Komodakis, 2016) (applied on CIFAR100 datasets) and VGG variants (Simonyan & Zisserman, 2014) (applied on CIFAR10 dataset). Both architectures use batch normalization layers (Ioffe & Szegedy, 2015) since they greatly facilitate the training procedure.

In order to build a set of models with a wide range of generalization performances, we vary hyperparameters that are known to be critical. Since varying multiple hyperparameters improves the identification of causal relationships, *we vary 8 different hyperparameters*: learning rate, batch size, optimizer (SGD or Adam (Kingma & Ba, 2015)), weight decay, dropout rate (Srivastava et al., 2014), data augmentation, network depth and width. In total, we elaborated a set of 192 hyperparameter configurations for each dataset-architecture pair[4]. More information about the hyperparameter configurations, training procedures and the resulting model performances are provided in Appendix (A.1, A.2, A.3 and A.4). It is worth noting that most models achieve close to $100\%$ training accuracy. Hence, models with poor test accuracy largely overfit.

---

[3]Activating $25\%$ of the samples was an arbitrary choice that we did not seek to optimize. Studying the influence of this hyperparameter on the results is left as future work.

[4]The final set of VGG variants trained on CIFAR10 results from only 144 hyperparameter configurations as the network could not reach $\sim 100\%$ training accuracy with high dropout rates.

## 3.2 EVALUATING THE MEASURES' PREDICTIVE POWER

Jiang et al. (2020) provides multiple criteria to evaluate generalization measures. We opted for the granulated Kendall coefficient for its simplicity and intuitiveness. This coefficient compares two rankings of the models, respectively provided by (i) the generalization measure of interest and (ii) their actual generalization performance. The Kendall coefficient is computed across variations of each hyperparameter independently. The average over all hyperparameters is then computed for the final score. The goal of this approach is to better capture causal relationships by not overvaluing measures that correlate with generalization only when specific hyperparameters are tuned.

We compare our intraclass clustering-based measures to sharpness-based measures. The latter constituted the most promising measure family from the large-scale study presented in Jiang et al. (2020). Among the many different sharpness measures, we leverage the magnitude-aware versions that measure sharpness through random and worst-case perturbations of the weights (denoted by $\frac{1}{\sigma'}$ and $\frac{1}{\alpha'}$, respectively, in Jiang et al. (2020)). We also include the application of these measures with perturbations applied on kernels only (i.e. not on biases and batch normalization weights) with batch normalization layers in batch statistics mode (i.e. not in inference mode). We observed that these alternate versions often provided better estimations of generalization performance. We denote these measures by $\frac{1}{\sigma''}$ and $\frac{1}{\alpha''}$.

## 4 RESULTS

This section starts with a thorough evaluation of the relationship between the four proposed measures and the generalization performance, using the setup described in 3. Then, it presents a series of experiments to better characterize intraclass clustering, the phenomenon we expect to be captured by the measures. These experiments include (i) an analysis of the measures' evolution across layers and training iterations, (ii) a study of the neuron-level measures' sensitivity to $k$ in the mean over top-$k$ operation, as well as (iii) visualizations of subclass extraction in individual neurons.

### 4.1 EVALUATING THE MEASURES' RELATIONSHIPS WITH GENERALIZATION

We compute all four measures on the models trained on the CIFAR100 superclasses, and the two variance-based measures on the models trained on standard CIFAR100 and CIFAR10. We set $k = 30$ for the neuron-level measures, meaning that 30 neurons per subclass (for $c_1$) or class (for $c_3$) are used to capture intraclass clustering. For the layer-level measures, we set $k = 5$ for residual networks and $k = 1$ for VGG networks.

We start our assessment of the measures' predictive power by visualizing their relationship with generalization performance in Figure 3. *We observe a clear trend across datasets, network architectures and measures, suggesting a tight connection between intraclass clustering and generalization.* To further support the conclusions of our visualizations, we evaluate the measures' predictive power through the granulated Kendall coefficient (cfr. Section 3.2). Tables 1, 2 and 3 present the granulated Kendall rank-correlation coefficients associated with intraclass clustering and sharpness-based measures, for the three dataset-architecture pairs (Tables 2 and 3 are in Appendix A.5).

The Kendall coefficients further confirm the observations in Figure 3 by revealing strong correlations between intraclass clustering measures and generalization performance *across all hyperparameters*. In terms of overall score, intraclass clustering measures surpass the sharpness-based measures variants by a large margin across all dataset-architecture pairs. On some specific hyperparameters, sharpness-based measures outperform intraclass clustering measures. In particular, $\frac{1}{\alpha'}$ performs remarkably well when the batch size parameter is varied, which is coherent with previous work (Keskar et al., 2017).

### 4.2 INFLUENCE OF $k$ ON THE KENDALL COEFFICIENTS OF NEURON-LEVEL MEASURES

In our evaluation of the measures in Section 4.1, the $k$ parameter, which controls the number of highest values considered in the mean over top-$k$ operations, was fixed quite arbitrarily. Figure 4 shows how the Kendall coefficient of $c_1$ and $c_3$ changes with this parameter. We observe a relatively low sensitivity of the measures' predictive power with respect to $k$. In particular, in the case of

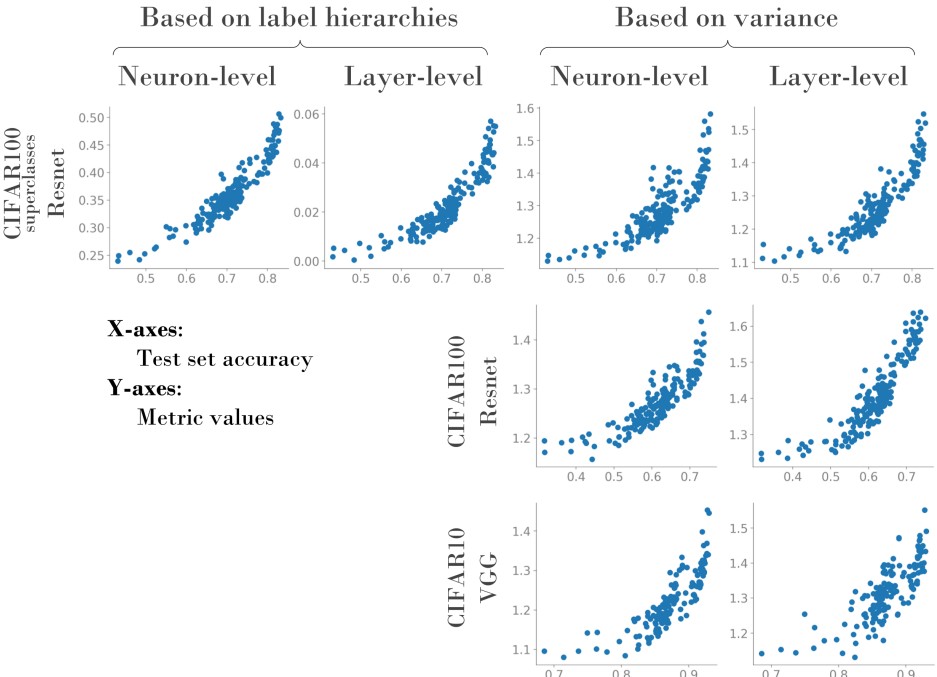

Figure 3: Visualization of the relationship between the four proposed intraclass clustering measures and generalization performance, across datasets and network architectures. The four columns correspond to $c_1$, $c_2$, $c_3$ and $c_4$ measures respectively. All measures display a tight connection with generalization performance, suggesting a crucial role for intraclass clustering in the implicit regularization of deep neural networks.

Table 1: Kendall coefficients for resnets trained on CIFAR100 superclasses. Overall, the intraclass clustering measures surpass the sharpness-based measures by a large margin.

|  |  | learning rate | batch size | weight decay | optim. | dropout rate | data augm. | width | depth | total score |
|---|---|---|---|---|---|---|---|---|---|---|
| Intraclass clustering | $c_1$ | 0.88 | 0.31 | 0.38 | 0.67 | 0.96 | **1.0** | **0.81** | 0.69 | 0.71 |
|  | $c_2$ | 0.86 | 0.5 | **0.67** | 0.58 | **0.99** | **1.0** | 0.38 | 0.62 | 0.7 |
|  | $c_3$ | 0.88 | 0.6 | 0.46 | 0.62 | 0.81 | **1.0** | **0.81** | 0.66 | **0.73** |
|  | $c_4$ | **0.89** | 0.69 | 0.62 | 0.65 | 0.86 | **1.0** | 0.44 | 0.69 | **0.73** |
| Sharpness | $1/\sigma'$ | 0.81 | 0.51 | 0.31 | 0.69 | 0.28 | -0.58 | 0.67 | 0.61 | 0.41 |
|  | $1/\sigma''$ | 0.86 | 0.58 | 0.17 | 0.4 | -0.05 | 0.42 | 0.69 | **0.72** | 0.47 |
|  | $1/\alpha'$ | 0.88 | **0.94** | 0.29 | 0.26 | 0.6 | 0.08 | -0.03 | -0.09 | 0.37 |
|  | $1/\alpha''$ | 0.85 | 0.8 | 0.48 | **0.71** | 0.16 | -0.08 | 0.08 | 0.34 | 0.42 |

Resnets trained on CIFAR100 the Kendall coefficient associated with $c_3$ seems to stay above $0.7$ for any $k$ in the range $[1, 900]$. The optimal $k$ value changes with the considered dataset and architecture. We leave the study of this dependency as a future work.

Observing the influence of $k$ also confers insights about the phenomenon captured by the measures. Figure 4 reveals that very small $k$ values work remarkably well. Using a single neuron per subclass ($k = 1$ in Equation 1) confers a Kendall coefficient of $0.69$ to $c_1$. Using a single neuron per class confers a Kendall coefficient of $0.78$ to $c_3$ in the case of VGGs trained on CIFAR10. *These results suggest that individual neurons play a crucial role in the extraction of intraclass clusters during training.* The fact that the Kendall coefficients monotonically decrease after some $k$ value suggests that the extraction of a given intraclass cluster takes place in a sub part of the network, indicating some form of specialization.

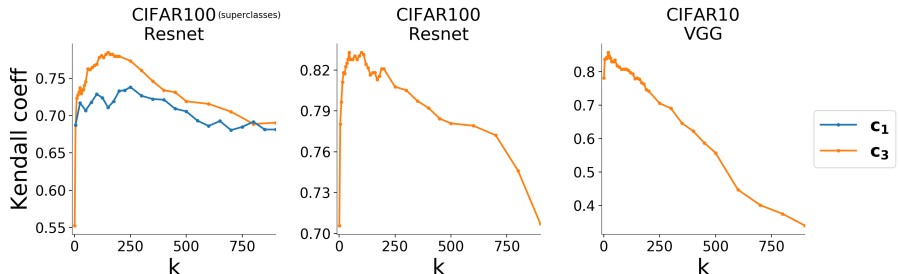

Figure 4: Plots showing how the Kendall coefficients of $c_1$ and $c_3$ change with parameter $k$ (cfr. Equations 1 and 3). The $k$ parameter associated to $c_1$ is multiplied by $5$ in the plots, to enable comparison with $c_3$ (there are $5$ subclasses in each of CIFAR100's superclasses). The total number of neurons varies from $1920$ to $2880$ in Resnets and from $960$ to $1440$ in VGGs. The plots reveal that generalization performance can be quite accurately estimated using the representations of a surprisingly small set of neurons ($k = 1$, i.e. a single neuron per class, suffices in some cases).

## 4.3 EVOLUTION OF THE MEASURES ACROSS LAYERS

We pursue our experimental endeavour with an analysis of the proposed measures' evolution across layers. For each dataset-architecture pair, we select $64$ models which have the same depth hyperparameter value. We then compute the four measures on a layer-level basis (we use the top-5 neurons of each layer for the neuron-level measures) and average the resulting values over the $64$ models. Figure 5 depicts how the average value of each measure evolves across layers for Resnets trained on CIFAR100 superclasses. The results associated with the two other dataset-architecture pairs are in Appendix A.5, Figures 9 and 10.

We observe two interesting trends. First, all four measures tend to increase with layer depth. *This suggests that intraclass clustering also occurs in the deepest representations of neural networks, and not merely in the first layers, which are commonly assumed to capture generic or class-independent features.* Second, the variance based measures ($c_3$ and $c_4$) decrease drastically in the penultimate layer. We suspect this reflects the grouping of samples of a class in tight clusters in preparation for the final classification layer (such behaviour has been studied in Kamnitsas et al. (2018); Müller et al. (2019)). The measures $c_1$ and $c_2$ are robust to this phenomenon as they rely on relative distances inside a single class, irrespectively of the representations of the rest of the dataset.

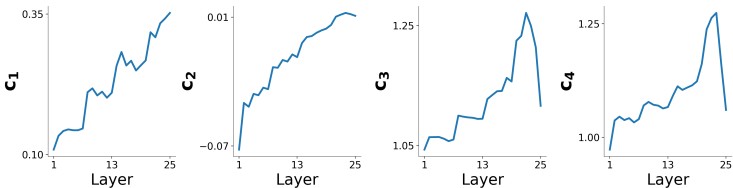

Figure 5: Evolution of each measure (after averaging over 64 models) across layers. The overall increase of the measures with layer depth suggests that intraclass clustering occurs even in the deepest representations of neural networks.

## 4.4 EVOLUTION OF THE MEASURES OVER THE COURSE OF TRAINING

In this section, we provide a small step towards the understanding of the dynamics of the phenomenon captured by the measures. We visualize in Figure 6 the evolution of the measures over the course of training of three Resnet models. The first interesting observation comes from the comparison of models with high and low generalization performances. It appears that *their differences in terms of intraclass clustering measures arise essentially during the early phase of training*. The second observation is that significant increases in intraclass clustering measures systematically coincide with significant increases of the training accuracy (in the few first epochs and around epoch 150,

where the learning rate is reduced). This suggests that supervised training could act as a necessary driver for intraclass clustering ability, despite not explicitly targeting such behaviour.

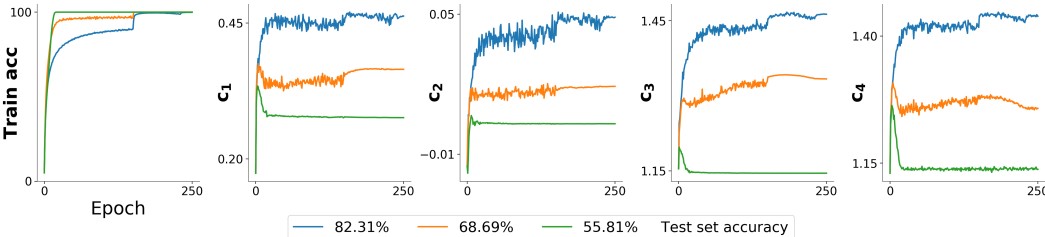

Figure 6: Evolution of the intraclass clustering measures over the course of training for three models with different generalization performances. We observe that the differences between models with high and low generalization performance arise essentially in the early phase of training.

### 4.5 VISUALIZATION OF SUBCLASS EXTRACTION IN INDIVIDUAL NEURONS

We have seen in Section 4.2 that the measure $c_1$ reaches a Kendall coefficient of $0.69$ when considering a single neuron per subclass ($k = 1$ in Eq. 1). Visualizing the training dynamics in this specific neuron should enable us to directly observe the phenomenon captured by $c_1$. We study a Resnet model trained on CIFAR100 superclasses with high generalization performance ($82.31\%$ test accuracy). For each of the 100 subclasses, we compute the selectivity value and the index of the most selective neuron based on the part of Eq. 1 to which the median operation is applied. We then rank the subclasses by their selectivity value, and display the training dynamics of the neurons associated to the subclasses with maximum and median selectivity values in Figure 7.

The evolution of the neurons' preactivation distributions along training reveals that the 'Rocket' subclass, which has the highest selectivity value, is progressively distinguished from its corresponding superclass during training. *The neuron behaves like it was trained to identify this specific subclass although no supervision or explicit training mechanisms were implemented to target this behaviour.* The same phenomenon occurs to a lesser extent with the 'Ray' subclass, which has the median selectivity value. We observed that very few subclasses reached selectivity values as high as the 'Rocket' subclass (the distribution of selectivity values is provided in Figure 11 in Appendix A.5). We suspect that the occurrence of such outliers explain why the median operation outperformed the mean in the definition of $c_1$ and $c_2$.

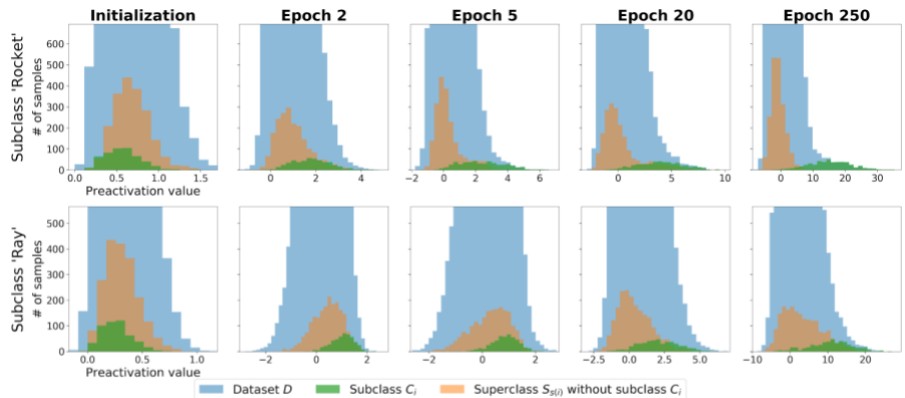

Figure 7: Evolution along training of the preactivation distributions associated with the neurons that are the most selective (cfr. Eq. 1) for 'Rocket' and 'Ray' subclasses. The neurons behave like they were trained to identify these specific subclasses although no supervision or explicit training mechanisms were implemented to target this behaviour.

### 4.6 DISCUSSION

Our results show that all four measures (i) strongly correlate with generalization, (ii) tend to increase with layer depth and (iii) change mostly in the early phase of training. These similarities suggest that the four measures capture one unique phenomenon. Since all measures quantify to what extent a neural network differentiates samples from the same class, the captured phenomenon presumably consists in the identification of intraclass clusters by neural networks. This hypothesis is further supported by the neuron-level visualizations provided in Section 4.5. Overall, our results thus provide empirical evidence for this works hypotheses, i.e. that intraclass clustering emerges during standard neural network training and constitutes an implicit form of regularization.

The mechanisms that induce intraclass clustering in deep neural networks remain undisclosed, leaving a lot of room for future work. In particular, Section 4.2 reveals that understanding the local nature of intraclass clustering could lead to better estimation of the optimal $k$ parameter and increase the measures' predictive power. Additionally, the neuron-level visualizations in Figures 2 and 7 suggest that quantifying the bimodal shape of a neuron's preactivations for each class might provide an alternative measure of intraclass clustering that doesn't require hierarchical labels.

## 5 RELATED WORK

This work follows the line of inquiry initiated by Neyshabur et al. (2015); Zhang et al. (2017), and tries to unveil the phenomena underlying the generalization ability of deep neural networks. We follow an empirical approach based on the design of measures that correlate with generalization performance. This approach has been extensively used in recent years, leading to measures such as sharpness of minima (Keskar et al., 2017), sensitivity (Novak et al., 2018), reliance on single directions (Morcos et al., 2018) and stiffness (Fort et al., 2019). To our knowledge, we are the first work to empirically study the relationship between intraclass clustering measures and generalization.

Many observations made in our paper are coherent with previous work. In the context of transfer learning, Huh et al. (2016) shows that representations that discriminate ImageNet classes naturally emerge when training on their corresponding superclasses, suggesting the occurrence of intraclass clustering. Sections 4.2 and 4.5 suggest a key role for individual neurons in the extraction of intraclass clusters. This is coherent with the large body of work that studied the emergence of interpretable features in the hidden neurons (or feature maps) of deep nets (Zeiler & Fergus, 2014; Simonyan et al., 2014; Yosinski et al., 2015; Zhou et al., 2015; Bau et al., 2017). In Section 4.4, we notice that intraclass clustering occurs mostly in the early phase of training. Previous works have also highlighted the criticality of this phase of training with respect to regularization (Golatkar et al., 2019), optimization trajectories (Jastrzebski et al., 2020; Fort et al., 2020), Hessian eigenspectra (Gur-Ari et al., 2018), training data perturbations (Achille et al., 2019) and weight rewinding (Frankle et al., 2020a;b). Morcos et al. (2018); Leavitt & Morcos (2020) have shown that class-selective neurons are not necessary and might be detrimental for performance. This is coherent with our observation that neurons that differentiate samples from the same class improve performance.

## 6 CONCLUSION

Our work starts from the observation that classes of samples from standard image classification datasets are usually structured into multiple clusters. Since learning structure in the data is commonly associated to generalization, we ask if state-of-the-art neural network extract such clusters. While no supervision or training mechanisms are explicitly programmed into deep neural networks for targeting this behaviour, it is possible that this learning ability implicitly emerges during training. Such a hypothesis is especially compelling as recent work on generalization conjectured the emergence of implicit forms of regularization during deep neural network training.

Our work provides four different attempts at quantifying intraclass clustering. We then conduct a large-scale experimental study to characterize intraclass clustering and its potential relationship with generalization performance. Overall, the results suggest a crucial role for intraclass clustering in the regularization of deep neural networks. We hope our work will stimulate further investigations of intraclass clustering and ultimately lead to a better understanding of the generalization properties of deep neural networks.

## ACKNOWLEDGMENTS

Thanks to the reviewers for suggesting several experiments that were added during the rebuttal phase and increased the quality of this paper.

Thanks to Amirafshar Moshtaghpour, Anne-Sophie Collin, Antoine Vanderschueren, Victor Joos de ter Beerst, Tahani Madmad, Pierre Carbonnelle, Vincent Francois, Gilles Peiffer for proofreading and helpful feedback.

Thanks to the Reddit r/machinelearning community for keeping us up to date with our fast-moving field.

Simon and Christophe are Research Fellows of the Fonds de la Recherche Scientique  FNRS, which provided funding for this work.

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

## A    APPENDIX

### A.1    Hyperparameter configurations

We define here the hyperparameter configurations used to build our set of models (cfr. Section 3.1). A total of 8 hyperparameters are tuned: learning rate, batch size, optimizer, weight decay, dropout rate, data augmentation, network depth and width. A straightforward way to generate configurations is to specify values for each hyperparameter independently and then generate all possible combinations. However, given the amount of hyperparameters, this quickly leads to unrealistic amounts of models to be trained.

To deal with this, we decided to remove co-variations of hyperparameters whose influence on training and generalization is suspected to be related. More precisely, we use weight decay only in combination with the highest learning rate value, as recent works demonstrated a relation between weight decay and learning rate (van Laarhoven, 2017; Zhang et al., 2019). We also don't combine dropout and data augmentation, as the effect of dropout is drastically reduced when data augmentation is used. Finally, we do not jointly increase width and depth, to avoid very large models that would slow down our experiments.

The resulting hyperparameter values are as follows:

1. **(Learning rate, Weight decay)**: $\{(0.01, 0.), (0.32, 0.), (0.1, 0.), (0.1, 4 \cdot 10^{-5})\}$
2. **Batch size**: $\{100, 300\}$
3. **Optimizer**: $\{SGD, Adam\}$
4. **(Dropout rate, Data augm.)**: $\{(0., true), (0., false), (0.2, false), (0.4, false)\}$
5. **(Width factor, Depth factor)**: $\{(\times1., \times1.), (\times1.5, \times1.), (\times1., \times1.5))\}$

We generate all possible combinations of these hyperparameter values (or pairs of values), leading to 192 configurations. Since dropout rates of $0.4$ lead to poor training performance on VGG variants, only $144$ configurations are used in these cases.

### A.2    Further information on the hyperparameters

It has been shown that different optimizers may require different learning rates for optimal performance (Wilson et al., 2017). In our experiments, we divide the learning rate by $100$ when using $Adam$ to improve its performance (the same approach is used in Jiang et al. (2020)). When using $SGD$ we apply a momentum of $0.9$.

We use the following data augmentations: horizontal and vertical shifts of up to 4 pixels, and horizontal flips. We fill the resulting image through reflection. This data augmentation procedure is widely used for the CIFAR datasets.

We use two types of network architectures: VGG and Wide ResNets. We build these architectures using four stages, separated by 3 pooling layers. This enabled a wider range of generalization performances, as the standard three stage versions of these architectures tend to be less capable of overfitting. Width factors of $\times1.$ and $\times1.5$ correspond to widths of 32 and 48 respectively in the layers of the first stage. The width of layers is doubled after every pooling layer. Depth factors of $\times1.$ and $\times1.5$ correspond to 9 and 13 layers for VGG variants, and 18 and 26 layers for Wide ResNet variants.

### A.3    Further information on the training procedure

We train models for 250 epochs, and reduce the learning rate by a factor $0.2$ at epochs $150, 230, 240$. Training stops prematurely if the training loss gets smaller than $10^{-4}$.

## A.4 MODEL PERFORMANCES

Cfr. Figure 8.

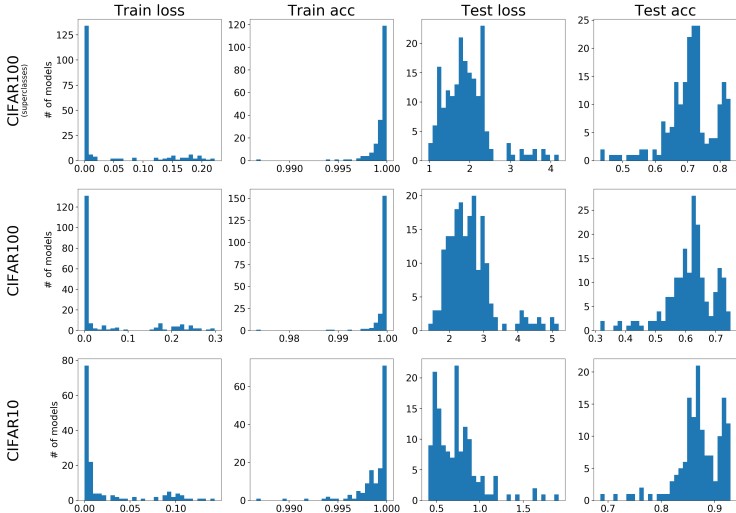

Figure 8: Histogram of performances of the set of models used in our experiments.

## A.5 SUPPLEMENTARY RESULTS

Table 2: Kendall coefficients for resnets trained on CIFAR100.

|  |  | learning rate | batch size | weight decay | optim. | dropout rate | data augm. | width | depth | total score |
|---|---|---|---|---|---|---|---|---|---|---|
| Intraclass | $c_3$ | 0.94 | 0.65 | **0.62** | 0.58 | **1.0** | **1.0** | **1.0** | **0.78** | **0.82** |
| clustering | $c_4$ | 0.93 | 0.62 | **0.62** | 0.21 | **1.0** | **1.0** | 0.91 | **0.78** | 0.76 |
| Sharpness | $1/\sigma'$ | 0.88 | 0.68 | 0.17 | **0.8** | 0.4 | -0.62 | 0.94 | 0.61 | 0.48 |
|  | $1/\sigma''$ | 0.92 | 0.61 | 0.12 | 0.35 | -0.06 | 0.31 | 0.94 | 0.53 | 0.47 |
|  | $1/\alpha'$ | **0.96** | **0.96** | 0.17 | 0.25 | 0.54 | 0.15 | -0.16 | -0.23 | 0.33 |
|  | $1/\alpha''$ | **0.96** | 0.91 | 0.42 | 0.64 | 0.12 | -0.25 | 0.17 | 0.14 | 0.39 |

Table 3: Kendall coefficients for VGG networks trained on CIFAR10.

|  |  | learning rate | batch size | weight decay | optim. | dropout rate | data augm. | width | depth | total score |
|---|---|---|---|---|---|---|---|---|---|---|
| Intraclass | $c_3$ | 0.92 | 0.83 | **0.67** | 0.51 | **0.92** | **1.0** | **1.0** | 0.88 | **0.84** |
| clustering | $c_4$ | 0.86 | 0.75 | 0.33 | 0.29 | **0.92** | **1.0** | 0.54 | 0.92 | 0.7 |
| Sharpness | $1/\sigma'$ | 0.86 | 0.62 | -0.25 | 0.6 | -0.04 | -0.27 | **1.0** | 0.85 | 0.42 |
|  | $1/\sigma''$ | 0.9 | 0.67 | 0.11 | **0.69** | **0.92** | 0.19 | **1.0** | **0.94** | 0.68 |
|  | $1/\alpha'$ | **0.94** | **0.89** | 0.61 | 0.53 | 0.67 | 0.77 | 0.15 | 0.06 | 0.58 |
|  | $1/\alpha''$ | 0.93 | 0.67 | 0.36 | 0.54 | 0.69 | -0.02 | 0.15 | -0.15 | 0.4 |

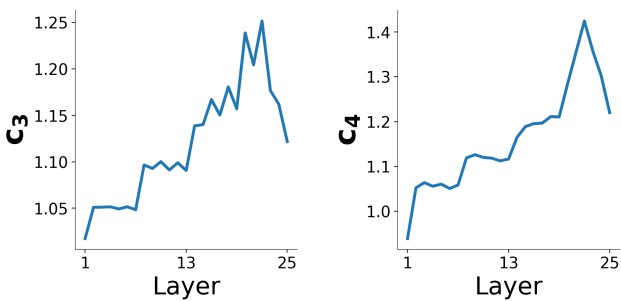

Figure 9: Evolution of the measures across layers for Resnets trained on CIFAR100

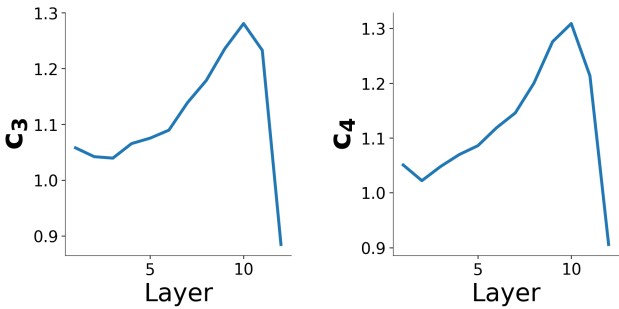

Figure 10: Evolution of the measures across layers for VGGs trained on CIFAR10

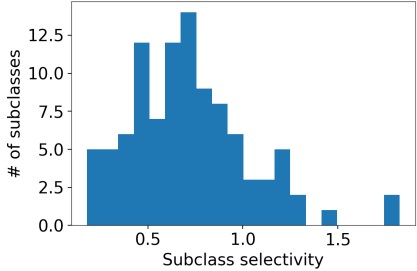

Figure 11: Distribution of neural subclass selectivity values (cfr. measure $c_1$) over the 100 subclasses of CIFAR100. For each subclass, neural subclass selectivity is computed based on the most selective neuron in the neural network (i.e. $k = 1$). We observe that 1) only a few subclasses reach high selectivity values and 2) the selectivity values vary much across subclasses. We suspect that the outliers with exceptionally high selectivity values cause the median operation to outperform the mean in the measures $c_1$ and $c_2$.

