# OpenReview forum: "Intraclass clustering: an implicit learning ability that regularizes DNNs"
_ICLR.cc/2021/Conference — ICLR 2021 Poster_

### Official Review · AnonReviewer3 · 2020-10-28
**Promising idea but needs more work**

**Rating:** 6
**Confidence:** 4

**Review:**

**Summary**:

This paper introduces a notion coined “intraclass clustering” that describes a deep neural network’s implicit ability to cluster within data. 4 different quantities that measure the networks’ clustering ability are proposed, and large scale experiments show that they are highly effective at predicting the models’ generalization ability across diverse types of hyperparameters.

**Pros**:
1. The paper presents an interesting idea that also seems to be practically highly relevant based on the experiments. Notably, it performs *nearly perfectly* on several interesting hyperparameters that are challenging for previous measures. I believe that there is definitely something unusual in this idea and it is worth future research.
2. The experiments on the evolution of the coefficients is extremely interesting, since this may indicate that we do not need to train the models to convergence to verify how good the models are.
The quantity is simple and seems to be efficient to compute (?)

**Cons**:
1. I think this paper is not very well-written and needs much more work. For example, why are there 4 different coefficients when two of them clearly outperform the previous two based on label hierarchies, which are nonetheless the motivations for the work and supposedly model the actual intraclass clustering ability? The paper claims that they are complementary, but the experiments, in my opinion, suggests otherwise. No further explanation is offered for this phenomenon, and I am just left wondering. I think this warrants much more investigation and explanation.
2. What is the exact definition of a neuron and a layer? The activation tensor is of shape NxHxWxC. Is the neuron a single scalar or single channel (NxHxWx1)? Likewise, is the layer the whole tensor of a single channel? This is not very clear in the paper, and it would help to spell out exactly how the variance or means are computed (using formula).
3. It would be nice to have a couple more comparisons instead of just sharpness.
4. It’s not immediately clear to me how this phenomenon can be converted to a learning theoretical argument about generalization, but this is minor.
5. I must admit that it is not clear to me why this phenomenon should capture generalization. I assume the authors had some conjectures or intuitions, so it would be nice to include that in the paper. Specifically, what is special about “intraclass”? Does this mean good models are just memorizing prototypes?
6. In 3.3.1, “If the standard deviation computed over the samples of a class is high compared to the standard deviation computed over the entire dataset, we infer that the neuron has learned features that differentiate samples belonging to this class”, why does standard deviation capture this? I think an justification would be nice. The figure in some sense does it but I don't think that's thorough enough.
7. It would be good to have analysis of which layer is usually picked when doing max or top k.

**Comments**:
1. Can intraclass clustering be viewed as some kind of specialization of different parts of the model or some kind of compression? I think more interesting things can be said about this.
2. What is a_l in eq 2?
3. For figure 4, is there adversarial noise for the green curve?
4. It would be interesting to see the evolution of the ranking instead of the coefficients as table 1-3 but this is quite expensive and I think the existing materials are already interesting enough.
5. It would be nice to see the mutual information based metric from Jiang et al, although this is not high on priority. If the authors are interested, they could use the data and code released by PGDL competition at NeurIPS which includes the implementation of mutual information metric.
6. Are all the models interpolating? Would be nice to see a visualization of training loss or accuracy.


**Conclusion**:

This paper presents something that I personally believe is very interesting, if not exciting. But, the paper needs quite a lot of works before it can be published. I want to emphasize that I believe the idea is extremely promising, but the presentation, the execution and thoroughness of analysis unfortunately just miss the mark. As such, I am inclined to reject the paper for now. I will increase my scores if the authors can address my concerns.

**================================== Update after rebuttal ==================================**

I appreciate the reviewers' hard work for adding this many new materials in a short period of time. A large number of my concerns have been addressed and the quality of paper has improved significantly. Some newly added experiments give a lot more insights than the original draft. As such, I am in favor of accepting the paper and have increased my scores from 4 to 6.

However, I still have a few lingering questions in the light of the rebuttal and would love to see them addressed should the paper be accepted:
1. Re: figure 4, the question is why the performance of the green curve is so bad. I was wondering if some of the labels are wrongly labeled to induce this effect.
2. For reducing the feature map to a single quantity, why is the max operation chosen? Intuitively, it makes more sense to use mean or even order statisitc to make the metric more robust.

I also understand that many of the experiments cannot be added in a short period of time and hope that the authors add them as promised in the rebuttal.

---

> ### Author Response · Authors · 2020-11-20
> **Reply to the reviewer's questions and suggestions (Part 1)**
>
> We thank the reviewer for the positive comments and constructive feedback. We’ve put lots of efforts to integrate the comments, and are convinced that our paper gained much value during this process. We now detail our replies to the "cons" and "comments" made by the reviewer.
>
> **Cons**:
> 1. Beyond considering intraclass clustering at the neuron (C1 and C3) or layer (C2 an C4) level, the metrics studied in the paper complement each other for the following reasons.
> On the one hand, metrics C1 and C2, which are based on label hierarchies, have the advantage to offer an explicit definition of sub-classes but have the drawback to not capture the impact of arbitrary clusters. Hence, they rely on the assumption that human subclass labels approximately correspond to the clusters perceived by the network. This assumption might be partly valid, but it is also reasonable to expect that other clusters make more sense for a neural network.
> On the other hand, metrics C3 and C4 have the advantage not making any prior assumption about the composition of potential intraclass clusters, but do not allow to consider them individually as for C1 and C2.
> Our goal was thus to design different measures, quantifying differently the same intraclass clustering phenomenon. Each measure adopts a different perspective with respect to the phenomenon, as stated in the measure definitions. The fact that they capture the same phenomenon is suggested by the fact that they all lead to similar results/conclusions. To clarify this argument, this discussion has been added to Section 4.6.
> Please also note that the performances of C1 and C2 in terms of Kendall coefficients have increased in the new version of the paper, by the use of mean over top-k instead of maximum operation. Mean over top k was already used by measures C3 and C4.
>
> 2. We agree that the concept of neuron requires precisions in the context of convolutional layers. We added the following clarifications to the paper: “We call pre-activations the values preceding the application of the ReLU activation function and activations the values following it. In convolutional layers, a neuron refers to an entire feature map. The spatial dimensions of such a neuron's (pre-)activations are reduced through a global max pooling operation before applying our measures.”
>
> 3. We couldn’t add comparison to other measures in the rebuttal’s limited timeframe.
>
> 4. This is a complex endeavor. We hope our work will stimulate this in future works.
>
> 5.  First, as now more explicitly stated in our Related work section, several previous works inspired our work’s hypotheses. [1] shows that representations that discriminate ImageNet classes naturally emerge when training on their corresponding superclasses, suggesting the occurrence of intraclass clustering. [2][3] have shown that class-selective neurons are sometimes detrimental for performance, suggesting that neurons should differentiate samples from the same class to improve performance.
> Additionally, we present in the Introduction and Conclusion of our paper the following argument. Our work starts from the observation that classes of samples from standard image classification datasets are usually structured into multiple clusters. Since learning structure in the data is commonly associated to generalization, the extraction of such clusters might be beneficial for generalization performance.
> [1] What makes ImageNet good for transfer learning?, Huh et al., 2016
> [2] On the importance of single directions for generalization, Morcos et al., ICLR 2018
> [3] Selectivity considered harmful: evaluating the causal impact of class selectivity in DNNs, Leavitt et al., 2020
>
> 6. As stated in the paper, we hypothesize that the discrimination of intraclass clusters should be reflected by a high variance in the representations associated to a class. If all the samples of a class are mapped to close-by points in the neuron- or layer-level representations, it is likely that the neuron/layer did not identify intraclass clusters.
> Additionally, our work also provides an experimental argument. Indeed, measure c_3 (the one based on the standard deviations) has the same behaviour as measures c_1 and c_2 in terms of correlation with generalization, evolution across layers (which is a new result) and evolution across training iterations. We believe these similarities indicate that c_3 captures the same phenomenon as c_1 and c_2. Since all measures quantify to what extent a neural network differentiates samples from the same class, the captured phenomenon presumably consists in the identification of intraclass clusters by neural networks. This hypothesis is further supported by the neuron-level visualizations provided in Figure 7 (which is a new result). We added this discussion in Section 4.6 of our paper.

---

> ### Author Response · Authors · 2020-11-20
> **Reply to the reviewer's questions and suggestions (Part 2)**
>
> 7. We’ve added an experiment to study the evolution of the measures across layers (Section 4.3). Since the measures gradually increase with layer depth, these results suggest that intraclass clustering does not only happen in the first layers (which are known to learn generic and class-independent features). Moreover, the fact that this trend occurs for all four measures supports the idea that they all capture the same phenomenon. We thank the reviewer for this very relevant proposal.
>
> **Comments**:
>
> 1. We now provide additional results in Section 4.2 (Figure 4) inspecting how the number neurons considered influences the measures c_1 and c_2. As discussed in this section, the results tend to indeed confirm that  the identification of a cluster takes place in a sub part of the network, through some form of specialization. This specialization is further confirmed by the visualization of the extraction of subclasses by individual neurons, now presented in Section 4.5. Thank you for this comment, which encouraged us to run these additional experiments.
>
> 2. It denotes the activation in the $l^{th}$ layer. This has been clarified in the paper
>
> 3. We did not understand this comment. Could you please elaborate?
>
> 4. We presume that the reviewer would be interested in visualizing how the ranking associated to our proposed measures evolves along training. We are short in time during this rebuttal to provide such an analysis. However, from the evolution of the measures depicted in Figure 6, we anticipate that the ranking resulting from our measures rapidly converges to its final state.
>
> 5. Again, we were too short in time to compute the mutual information-based metric. In case of acceptance, this should not be a problem to include it in the final version of our paper.
>
> 6. Indeed, most models reach very close to 100% training accuracy. The distribution of training accuracies is now presented in Figure 8 (Appendix 4), together with training loss, testing loss and testing accuracy distributions.

---

### Official Review · AnonReviewer4 · 2020-10-28
**Review - Interesting insights**

**Rating:** 7
**Confidence:** 3

**Review:**

Summary:
This work investigates whether intra-class separation in the latent space of a neural network correlates with its generalization capabilities. To perform this, the authors design 4 measures that attempt to capture whether separate sub-clusters are formed for different sub-classes of a (super-)class. All measures take into account all activations in all layers of network. 2 measures operate at neuron level, 2 at layer level. 2 are designed for cases where we do have explicit sub-class labels, and 2 are for the case that no such labels are given. The work performs multiple experiments on Cifar10 & Cifar100 & Cifar100(superclasses) where multiple models are trained with varying hyper-parameter configurations, each leading to different performance. Then, it is shown that the 4 measures correlate positively with performance, which according to the authors suggests that intra-class separation within the network is happening and is important for generalization.

------------------------------------------------------------
Reasons for score:
I am very borderline on this. On one hand it is interesting to see such a strong correlation of the intra-class separation with generalization, but on the other hand there are many questions that the study leaves unanswered regarding whether we are measuring what we think we are measuring, whether the results are sensitive or not to choice of hyperparameters for the measures, etc. In general, I am currently left quite unconfident in drawing strong conclusions. Perhaps the rebuttal will help.

------------------------------------------------------------

Pros:

1) The main research questions, whether the networks have an inductive bias to learn features that perform intra-class separation, without a training loss that demands it, and whether this separation actually correlates with generalization, are interesting questions.

2) The results suggest a positive correlation of intra-class separability with generalization. This may be an interesting finding to parts of the community. For example there is work that attempts the opposite to improve generalization, introducing losses for learning a single tight cluster per class (losing intra-class separability) and show that this improves generalization. Contradicting evidence could spark interesting discussion / research in the community. (see references [1, 2] further down in my comments)

3) There is a very significant number of experiments performed (500+), which may offer significant information to the community.

------------------------------------------------------------

Cons:


1) The work presents no analysis of the behaviour of the measures with respect to design choices or the hyperparameters of the measures. As a result, I think we do not get much insight in “exactly” what is being measured. For example, are the measures capturing intra-classs eparation at 1st or last layer? Are the results robust to hyperparameters? Insights like this are important to understand the value of the results. The empirical investigation is unfortunately limited to simply reporting a seemingly high correlation of the presented metrics with the generalization, without further insights. This limited insight is particularly troublesome when coupled with several questions I have about the design of the measures and their configuration, which leave me unconfident about the rigor of the study, and whether we are indeed measuring intra-class separation in meaningful ways. (see below “Questions for rebuttal” for more details)

2) There are design choices with respect to the measures and their application on the net that seem adhoc, without appropriate explanation. (why use k-top values and not all? Why normalize the pre-activations and take 25% top percent? And so on so forth. See detailed questions below)

3) Similarly, there are configuration parameters that the work merely mentions the value chosen (e.g. k=5), without any explanation how this value was chosen, nor any sensitivity study of whether results would change for different values. This adds the unconfidence I have about the rigorousness of the study.

4) Reproducibility seems low. Code is not available, there are no details on the values explored for the hyperparameters of the networks in the main experiments, such as learning rates, etc, and it seems challenging for another party to re-run 500+ experiments to reproduce the results. It is especially in such studies that rigorous analysis for convincing results is extra important, which I think leaves a lot to be desired here.

5) Experiments are performed only on CIFAR 10 / 100. I understand that performing all these experiments is expensive, but on the other hand, results are limited only to 1 type of data. Do we know if conclusions generalize?

------------------------------------------------------------

Questions to address during rebuttal period:

In Sec 3.1, please clarify explicitly whether “pre-activations” are before or after normalization by batch-norm. (this will inter-play with statistics used in the study such as sigma_n,D).

Sec. 3.2.2 and Sec.3.3.2: Why use the cosine distance instead of Euclidean? If I am note mistaken, Euclidean would also evaluate whether a cluster is tight (concentrated closer into a point), rather than only the angle that the cosine measures? I don’t see a reason why the magnitude of activations wouldn’t matter for cluster/class discrimination in an arbitrary deep net. Please clarify in text why you prefer the use of cosine and whether you believe things would be similar with other distances (e.g. Euclidean)

Why is Eq 1 and Eq 3 done on the “pre-activations”, while Eq 2 and Eq 4 are done on the “activations” (after ReLU)? Have Eq 2 & 4 been tried on pre-activations and vice versa? Perhaps this also relates to why Eq2&4 are on cosine distance instead of Euclidean. If so, state reasons explicitly, and if you ve tried things differently.

Why do Eq. 1 and Eq 2 use median_i and Eq 3, 4 use mean_i ? Any theoretical or empirical support to this? Does the choice influence results? Please explicitly state in the paper too.

Why do Eq 1 and Eq 2 use max_n and max_l respectively, while Eq 3 use mean_n and mean_l ? Any theoretical or empirical support to this? Does the choice influence results? Please explicitly state in the paper too.

The max_n and max_l operators, as well as the mean_^k_n and mean_^k_l operators, choose a single n/l or k-top n/l neurons/layers with separability from the whole network. It could be the very first layer, or the last. Do you have any insights on which layers are usually chosen? I think this would add great insights if we knew. If first or last layers are chosen, does it make a difference on the value of this intra-class separability? For example, intra-cluster separability at last layers will say much about classification itself. At the first layer, it says that there are filters that merely identify multiple colors for the same class (both white and black cats). It would be nice to know what actually is being measured. Perhaps adding a discussion, or a plot showing which layers are usually chosen would be very useful, and it would add confidence to the reader about exactly what is being measured and how to interpret the results/conclusions. This is needed especially because some measures are not “actually” measuring intra-class clustering (e.g. c3 & c4) but something else, that we “assume” indicates intra-class clutsering, hence makes interpretation ambiguous.

How was the choice of values (k=5 for resnet, k=1 for VGG) done in Sec. 3.3.2? Do these values influence the behaviour? Would they influence the conclusions? It would be nice to have a sensitivity study on such values?

Sec 3.3.2: “we found it helpful…25% of the samples are activated”: Please elaborate more on how this was found, why do you think it’s important, how was the value 25% defined, and how the results change if this is not performed? It would be nice to have empirical analysis of how this influences how the measure behaves and whether it changes any results.

Fig 2: This is helpful in understanding why c3 and c4 may indicate indirectly some intra-class separation in some cases. But, what if a class presents uni-modal distribution with large std? (and not bimodal as in the example?). Can you think of a way to provide more supporting evidence that c3,c4 capture intra-class separation? (here is where a more extensive analysis would have also been useful)

Sec. 5.2: How do the results with sharpness based measures you get (Tabl1,2,3) compare to those reported by Jiang 2020? With a quick look of mine, they seem quite close. Please confirm and discuss in your paper. If the results are similar, it adds confidence to the reader that your experimental setup is correct, replicating theirs.

The “sharpness” measures do not require the class labels, right? If this is correct, then their capability of predicting generalization should not be directly comparable with the measures in this study, which do require the class labels. The study does not do this directly, but I think an explicit comment should be added for the reader, so that they are aware of this significant difference.

Fig.3: Can you discuss whether you think the values returned by the measures (y-axis) suggest the existence of actual intra-class clustering, and support the main assumption of the paper? In fact, the values 0.01-0.08 of the silhouette score seem very low (Silhouette is -1 to +1, with 0 meaning overlapping clusters), right? Similarly, the 0.1-0.7 returned by c1 also seems small. E.g. imagine 2 gaussians with same std for the 2 clusters. If they wouldn’t overlap, the distance of their centers would be 6+ stds, right? Rough estimate. 0.1-0.7 suggests quite a strong overlap I think. Perhaps adding a discussion on this would be nice. If you indeed agree there is no strong supporting evidence of clear overlap, perhaps ensure there are no strong statements in the paper about it, and mostly emphasize on the correlation to generalization instead.

Sec 5.3 “strong increases of the training accuracy”: Can you explicitly state to which points (e.g. epoch) you are refering to? Moreover, what happened at epoch ~140 and caused the spike in training accuracy? Lowering learning rate? I would advise you explicitly state it.

Fig.4: How do you group all the models/experiments in Good/medium/low performance groups? Clarify in text. What exactly are these numbers (82.31%...) in the caption? State explicitly. The reader should not have to guess about anything.

What does Kendall Coefficient == 1.0 mean, when data-augmentation changes? That all measures predict absolutely perfectly generalization? Isn’t this weird? How do authors interpret this? Perhaps a short discussion would add to the reader’s understanding, insights, and confidence in the interpretation of the results.

There is a body of literature that designs losses for learning 1 single tight cluster per class, losing intra-class separation *in the last layers*, and shows this improves generalization of classifiers. The current study seems to contradict it, suggesting intra-class clusters help generalization. I think this should be discussed. The issue is that, because of no analysis, we don’t know whether the current measures find intra-class separation in early or late layers, to help us understand how the results connect exactly with the rest of the literature (contradicting it, or complementing it if only earlier layers are chosen by the max_l operators). This is one of the reasons that I think an analysis with further insights on how exactly the proposed measures compute is important (as I said previously), and add confidence to the reader by pointing out how they relate to existing literature.
If you would agree, here a couple of references that do tight clustering: A recent such work on deep nets I know is [1], which shows tight clustering improves generalization (the work is mainly on semi-supervision, but in the end they also have experiments for standard fully-supervised learning). Similarly, [2] show that label smoothing promotes tight-clustering and improves generalization (supervised learning).

[1] Kamnitsas et al, Semi-Supervised Learning via Compact Latent Space Clustering, ICML 2018

[2] Muller et al, When Does Label Smoothing Help? NeurIPS 2019.


To help with reproducibility: Is it possible to provide implementation of the measures (even if not the whole code is possible to release), and details about exactly what values have been explored for the network/training hyperparameters in the main experiments in Table1 etc? (e.g. in an appendix).

------------------------------------------------------------

Minor comments & some additional feedback for improving the work (which are not necessary to discuss within the rebuttal, but do address as many as possible):

I think Eq.2 is not well descriptive of what is being described in Sec. 3.2.2. Specifically, it shows that silhouete(a_l, C_i) is computed only on the set of a subclass C_i. If I understand correctly from the text, instead, you actually compute the mean silhoute score of all the samples in a SuperClass. So, I think it should be silhouette(a_l, S_s(i)) ?
Moreover, a_l is undefined. Is it supposed to be the cosine distances? Please clarify and update the text to make things clear. Perhaps give the equation for the silhouette score explicitly if this helps.

Sec 1. The current statement “could account for … invalidated” I think is too strong. There are many studies that showed that these help generalization (e.g. I think the positive effect of augmentation is unquestionable). I would suggest rephrasing it to something more tactful, like “not the sole factors for generalization performance” or something similar.

In Sec 3.3.1, I think that the range of appropriate values for k is dependent on the number of classes in the database. Perhaps you would like to state this explicitly. Consider if you could also derive a rule of thumb (likely empirically) of what is a good k with respect to number of classes.

Sec 4.2 “ranking of models…performance.”: I think this sentence descriing Kendall coeff could use a slight rephrase to be more intuitive.

Sec. 4.2: “by penalizing measures”: I am not sure that the approach “penalizes”. Rather, it seems because it takes average over loads of values, it merely is robust to outlier cases (measure + hyperparam tuned). Or, averages-away the influence of the hyperparam? Am I right? Perhaps a slight rephrase would avoid this confusion.

“is designed to better capture causality”: I would suggest this argument to be rephrased a bit to avoid interpretation that it accurately captures causality. It is merely robust (due to averaging across hyperparams) to the case of only 1 hyperparam correlating with generalization. Notice that Jiang et al 2020 also take special care to not over-state the capabilities of this measure to identify causal factors (see Sec 2.2.2 in their paper).


**============  Summary of improvements and revisiting reviewer’s score after rebuttal ==============**

Summary of main points of improvement related to my comments after the revision:

- The authors have significantly extended the analysis within the paper with more experiments to investigate the proposed measures and the deep nets’ behaviour in question. This adds a lot of value to the paper, offering more insights and support for the main claims.

- The authors have added many in-text clarifications about certain design choices (and improved some, e.g. by average k-max instead of max), which makes the study much clearer and adds confidence to the reader about interpretating the investigation and its results.

- The authors have added a study of the influence of the k parameter in the measures, which simultaneously offers confidence in the conclusions (conclusions hold for a significant range of values for k) and offers new insights about how separability happens (1 neuron vs multiple).

- Reproducibility is greatly improved, both by improved clarifications of the experimental settings within the text, and by providing the code in the supplementary.

- The authors improved discussions about the results with respect to related literature, taking into account and linking them also to papers that at first glance seem to be contradicting (e.g. references [1,2] I provided above that show tight clustering improves generalization). These links and discussions further improve my confidence in the conclusions.

Overall, the authors have addressed sufficiently most of my concerns, significantly improving the manuscript. The updated manuscript provides significantly more insights, which should be of interest to a part of the community. The extra analysis provides better support of the study’s hypothesis and claims. Remaining weaknesses of the paper include the CIFAR-only experimentation (we do not know if conclusions hold beyond it). I am happy to increase my review’s score from 5 (marginally bellow acceptance) to 7 (Good paper, accept).

= Minor =

In case the paper gets accepted, I would suggest the authors to try to complete Fig 4 with the layer-wise investigation, which they said was too expensive for the rebuttal period (I cannot estimate how expensive that can be. Hopefully it is possible in longer time period, and would complete nicely Fig 4).

---

> ### Author Response · Authors · 2020-11-20
> **Reply to the reviewer's questions and suggestions (Part 1)**
>
> We thank the reviewer for this extensive and very relevant feedback. We are convinced that the reviewer’s comments, by guiding the modifications we made during the rebuttal, significantly increased the quality of our paper.
>
> Before delving into the specific questions of the reviewer, we would like to start with a general note on the reviewer’s concerns with respect to the different design choices of our measures. We fully agree with the reviewer that the initial version of the paper completely failed to communicate the motivations/intuitions behind these choices. We regret this mistake and have put lots of energy in the rebuttal to address this issue. We thank the reviewer for underlying the necessity of completing our paper in that respect.
> Additionally, we would like to note that the need for many design choices is common to works around empirical measures of generalization. For example, we had to tune lots of parameters, mainly through trial and error, to reproduce the sharpness-based measures (e.g. the training parameters -optimizer, learning rate, nb_iterations-, epsilon, whether to use magnitude-aware variants). We even explored new design choices to further improve their performances (cfr. $\frac{1}{\sigma ''}$ and $\frac{1}{\alpha ''}$).  Jiang et al. 2020 uses a stopping criteria when building their set of models such that all models achieve similar training losses. We did not explore this design choice as this implies a considerable computational cost. We hope the reviewer understands that tuning the different design choices are part of the game and not a trick we used to embellish the performance of our own measures of generalization.
>
> We now delve into the specific questions of the reviewer. We use latin numerals to refer to each paragraph/question.
>
> (i)Pre-activations are after normalization. This has been clarified in our Terminology Section 2.1
>
> (ii)The main reason that led us to consider the cosine distance is the paper ‘Layer rotation: a surprisingly simple indicator of generalization in deep networks?’ presented at the ICML workshop ‘Identifying and Understanding Deep  Learning phenomena workshop’, http://deep-phenomena.org/. In this paper, the use of cosine distances successfully captures a link between generalization and the evolution of weight vectors along training. Since weight vectors and hidden representations are involved in the network through the same scalar products, cosine distance might also be appropriate to characterize the link between clustering of hidden representations and generalization. Since our preliminary experiments comparing Euclidian and cosine distances revealed that the cosine distance was slightly more conclusive regarding the clustering, we have focused our investigation on this metric. We’ve added this clarification to the paper.
>
> (iii)Measures c_1 and c_3 are defined at the neuron level. At this level, we are interested in evaluating whether the linear projection implemented by the neuron has been effective in splitting the data within a class. With respect to this purpose, the most natural feature lies in the moments of pre-activation values distributions. Hence, this is what we have reported in our paper. Our experiments have revealed that considering activation values instead of pre-activations for measure c_1 does not change the conclusions of our study, while this design choice was more crucial for measure c_3.
> In contrast, c_2 and c_4 are defined at the layer level, and are primarily interested in figuring out whether clusters are present in the hidden representations of a network.  Therefore, they consider the representations at the input of layers, i.e. the activation values.
> We rephrased the definition of the metrics to clarify this point.
>
> (iv)In general, the median filter is known to be more robust to outliers than the mean filter, while the mean is supposed to be more appropriate in presence of a reasonably uniform noise processes.
> Measures c_1 and c_2 exploit the subclass label knowledge. In the new version of the paper, we observe and discuss in Section 4.5 the fact that some subclasses behave like outliers, as they are exceptionally well clusterized in the network’s internal representations. A median filter enables getting rid of those outliers, and the extraction of more representative behaviour. We’ve included this argument in the paper.
> In contrast, c_3 and c_4 ignore subclass labels and manipulate standard deviation metrics that are associated to probability distributions that aggregate all the intraclass clusters of a class. The noise affecting those aggregated metrics can be considered as reasonably uniform across classes, which justifies the use of a mean filter.

---

> ### Author Response · Authors · 2020-11-20
> **Reply to the reviewer's questions and suggestions (Part 2)**
>
> (v)We’ve unified the definitions of the measures such that they all use the mean over top-k values instead of the max operator (we also updated the results accordingly). We added a Result section 4.2 where we study the influence of k on the measures’ Kendall coefficients.  Since k=1 corresponds to the max operator, this analysis reveals to what extent the design choices highlighted by the reviewer influences the results. We observe that this design choice is not critical for the conclusions of our work.
>
> (vi) We’ve added an experiment to study the evolution of the measures across layers (Section 4.3). Since the measures gradually increase with layer depth, these results suggest that intraclass clustering does not only happen in the first layers (which, as the reviewer rightly points out, are known to learn generic and class-independent features). Moreover, the fact that this trend occurs for all four measures supports the idea that they all capture the same phenomenon. We thank the reviewer for this very relevant proposal.
>
> (vii)Section 4.3 performs a sensitivity analysis of parameter k for the neuron-level measures. We did not include such an analysis for the layer-level measures, as these are more computationally expensive (c_4 especially).
>
> (viii)We believe this improves the measure by making it invariant to rescaling and translation of each neuron's preactivations. 25% was an arbitrary choice, and we did not try other values as it worked as is. We leave the study of this hyperparameter’s influence for future work. These comments have been added to the paper.
>
> (ix)Our main argument is that these measures have the same behaviour as measures c_1 and c_2 in terms of correlation with generalization, evolution across layers (which is a new result) and evolution across training iterations. We believe these similarities indicate that all four measures capture the same phenomenon. Since all measures quantify to what extent a neural network differentiates samples from the same class, the captured phenomenon presumably consists in the identification of intraclass clusters by neural networks. This hypothesis is further supported by the neuron-level visualizations provided in Figure 7 (which is a new result). We added this discussion in Section 4.6.
>
> (x)The results we provide have been reproduced from the method presented in Jiang 2020. As explained above, this has required some tuning effort. Similar to their work, sharpness measures struggle when regularization through noise is used in our models (e.g. dropout). Moreover, we observe that sharpness measures work very well for the batchsize parameter. This is also coherent with Keskar et al., 2017, which popularized the usage of sharpness measures to explain the generalization gap arising from large batch training. As suggested by the reviewer, we’ve added this comment to the paper.
>
> (xi) Sharpness measures do require class labels. Indeed, sharpness is based on the loss, which uses the true labels associated to each training sample.
>
> (xii)We’ve added in Section 4.5 a visualization to observe at neuron-level to what extent subclasses are effectively differentiated from their superclass during the training process. As noted by the reviewer, the overall measures, which are aggregated over all subclasses, are relatively low. However, we observe through this new visualization that some subclasses are progressively and significantly isolated from their superclass during the training process, which supports the existence of intraclass clustering.
>
> (xiii) We refer to the beginning of training and to he spike in training accuracy around epoch 150. This spike is indeed induced by a decrease of the learning rate. We’ve added these comments to the paper.
>
> (xiv)The plots correspond to three models, and not three groups of models. The numbers in the caption refer to the test accuracies of each of the three models. We’ve clarified this in the paper.
>
> (xv)A Kendall coefficient to one corresponds to a perfect prediction of the ranking of the models (not their exact generalization performance). Predicting the ranking is much easier. This is especially the case when considering data augmentation, whose application drastically increases generalization performance, leading to strong generalization disparities among models with different levels of data augmentation.

---

> ### Author Response · Authors · 2020-11-20
> **Reply to the reviewer's questions and suggestions (Part 3)**
>
> (xvi) We thank the reviewer for bringing up these interesting works. We’ve added a reference to these papers in section 4.3 where we study the evolution of the measures across layers. Our results are coherent with their observations, as our variance-based measures significantly drop in the last layer. The measures based on layer hierarchies remain high, which is also coherent as they rely on relative distances inside a single class, irrespectively of the representations of the rest of the dataset.
> As suggested by the reviewer, we’ve also significantly reorganized the Related work section, in order to better highlight the connections and points of convergence with previous works.
>
> (xvii) We’ve added details about the hyperparameter configurations, training procedures and model performances in appendix. We now also provide code in the supplementary material.
>
> Finally, we’ve also incorporated all minor comments of the reviewer, although we do not detail them here.

---

### Official Review · AnonReviewer2 · 2020-10-28
**This is a well done piece of work with insightful conclusions**

**Rating:** 8
**Confidence:** 3

**Review:**

### 1. Brief summary
The authors note that in classification tasks there typically exist within-class groups of similar images that are not explicitly encoded in the coarse class label -- they call this intraclass clustering. They hypothesize that the ability of DNNs to recognize these intraclass clusters without being explicitly told about that could correlate with generalization. They then proceed to verify this on a range of networks, architectures, and a large number of hyperparameter configurations. They take care to establish causality where possible. In addition, they show that the intraclass clustering can be detected with simple variance-based methods, and that it emerges early in training.

### 2. Strengths
* This is a super interesting question and I really like the paper overall.
* I appreciate that you looked at a large hypercube of hyperparameters to establish correlation with generalization
* I also like the care you put into establishing causality
* The fact that you tried a simple variance based measure is also really good, especially given that it is very predictive!

### 3. Weaknesses
I think this paper is really good, I have nothing much to point out here. Possible a large range of architectures and scaling up to ImageNet would be useful to establish that this scales all the way to very large data, but it is very good as is!

### 4. Related papers that you might like
[1] You cite Stiffness: A New Perspective on Generalization in Neural Networks by Stanislav Fort, Paweł Krzysztof Nowak, Stanislaw Jastrzebski, Srini Narayanan (https://arxiv.org/abs/1901.09491) as measuring an amount of class-specific clustering. In that paper in Figure 9 they show that stiffness is aware of the super-classes of CIFAR-10 and even their super-super-classes (animals, etc.), which seems relevant here too. Though there it goes the other way round -- the training is on subclasses, generating awareness of superclasses. Your results are I'd stay stronger than that.

[2] In talking about the sensitivity to early stages of training The Break-Even Point on Optimization Trajectories of Deep Neural Networks by Stanislaw Jastrzebski, Maciej Szymczak, Stanislav Fort, Devansh Arpit, Jacek Tabor, Kyunghyun Cho, Krzysztof Geras (https://arxiv.org/abs/2002.09572 and ICLR 2020) might be relevant, where they also establish a very strong effects of the early stages of training.

[3] Deep learning versus kernel learning: an empirical study of loss landscape geometry and the time evolution of the Neural Tangent Kernel by Stanislav Fort, Gintare Karolina Dziugaite, Mansheej Paul, Sepideh Kharaghani, Daniel Roy, Surya Ganguli (https://arxiv.org/abs/2010.15110 and NeurIPS 2020) also shows a very strong effect of the early stages of training on a large number of DNN measures.

### 5. Summary
This paper is really good! It starts with a strong hypothesis, verifies it on a large number of experiments, is mindful of causality and generates a potentially practically useful insight into generalization. Well done!

---

> ### Author Response · Authors · 2020-11-20
> **Thank you!**
>
> We thank the reviewer for these very positive comments! And we’ve been pleased to study the suggested papers. We’ve added references to these impressive works in the relevant paragraphs of our related work section.
>
> Let us also specify that numerous experiments have been added to the paper based on the other reviewers’ feedback. We believe these new results are of great interest and encourage the reviewer to take a look at them. They are also summarized in our general answer addressed to all reviewers.

---

### Official Review · AnonReviewer1 · 2020-10-28
**Interesting paper**

**Rating:** 6
**Confidence:** 4

**Review:**

This paper studies the intraclass clustering ability of neural networks trained in supervised learning and found that networks show intraclass clustering ability despite not explicitly enforced by the label to do so. And criterions based on those correlate well with model generalization performance.

However, I think the current state of the paper is still a bit below the standard of ICLR, and there are a few ways that this paper could be potentially improved.

1. I think observation 1 (clustering is learned without explicit constraints) is more interesting than observation 2 (correlate well with generalization performance), because it is intuitive that if you can classify a finer class assignment well, then you should also do well on more coarse classes. Maybe one thing that can be added to the experiments is to study whether the intraclass clusterability could also overfit. In other words, do you see a big generalization gap if you measure your criterions on the training set and test set separately?

2. As mentioned in the previous point, I think it is probably worth allocating more space to study how / why the intra-class clustering ability is learned during training despite supervised with corse label assignments. I find it especially intriguing from Fig.4 that all the clusterability metrics are actually very low at the beginning of the training where the network weights are random. My intuition was that at the beginning of training, there are probably some neurons that is lucky and separate the intra-class clustering well. But it seems that my intuition is wrong. It would be great if the paper has more experiments focusing on discovering what happens during the training that helps intraclass clustering. For example, are those intraclass clusters actually pushed away? Or they stay the same distances to each other, but benefits from by-product of supervised learning that shrinks all the clusters? Other experiments could be to study what kind of regularization / training techniques affect clusterability, and can we explicitly encourage clusterability to improve generalization during training, etc.

3. I'm a bit uncomfortable with the 'max' operation taken in the definition of clusterability criterions. It essentially searches over all the neurons looking for patterns that you want to see. Given that there are a lot of randomness in the neural network weights, this might not be depicting the 'typical' behavior of neurons. For example, it might be that there is a random neuron that happen to correlate very well with the intraclass clusterability. Maybe one can replace the max with the top_k averaging with a relatively large k like in the case of variances, or even with something like 99 percentile. A baseline can also be provided to see how much the randomness comes into play -- one can define a random clustering assignments, and compute the same criterion and see what number could be achieved.

4. Could you please include details in the experimental setup showing how do you get the intra-class clustering assignments that is needed to compute the criterions? I imagine for the 20-class CIFAR-100 you can just use the original labels as clustering assignment, but I did not find how you do it for other datasets.

5. Similar to the previous point, in practice, one usually do not have the intra-class cluster assignment available to compute those criterions. In contrast, the sharpness criterion can be computed without extra information. Can you add some materials to show what could be done in this case? Maybe show some way to approximately estimate the proposed metrics and study the estimation accuracies?

---

> ### Author Response · Authors · 2020-11-20
> **Reply to the reviewer's questions and suggestions**
>
> We thank the reviewer for this helpful feedback. We’ve incorporated many of the reviewer’s suggestions in our paper. We also make use of this rebuttal to clarify some misunderstandings that we detected in the reviewer’s comments (cfr. comments 4. and 5.).
>
> 1. Let us specify that observation 2 makes a link between fine class performance on the *training set* and coarse class performance on the *test set*. Fine class performance does not naturally imply coarse class performance, as these two performances are computed on different datasets (train and test).  This implication is even more surprising as all the models we study achieve close to 100% accuracy on the training set (this was not explicitly stated in the initial draft, and has been corrected). Hence, our work studies networks that overfit (i.e. whose performance does not transfer well from training to test set). Observation 2, by showing that intraclass clustering measures correlate well with generalization, reveals that intraclass clustering might reduce overfitting.  We believe this message is of interest to the community.
> It is indeed interesting to compare the values of the measures on the training and test sets. As such an experiment does not directly provide additional support to our work’s hypotheses, we did not include such an experiment in the paper.
>
> 2. Understanding how/why intra class clustering ability is learned is a complex and non-trivial research question. We hope that our paper, by the many intriguing questions it raises, will initiate discussion/collaboration between researchers with different backgrounds to create a collective effort.
> We agree with the reviewer that our paper should present more experiments to initiate this effort. Our initial submission already made a small step towards explaining the how, by studying the evolution of the measures over the course of training. We’ve now added numerous other experiments to complement this result. These experiments provide insights around the local nature of intraclass clustering (do different parts of the network specialize to classify different clusters? Cfr. Section 4.2) and in which layers it occurs (is it restricted to the first layers or do the deepest layers perform intraclass clustering too? Cfr. Section 4.3)
> Additionally, we added an experiment to specifically address one of the reviewer’s open questions: “are those intraclass clusters actually pushed away? Or they stay the same distances to each other, but benefits from by-product of supervised learning that shrinks all the clusters? “.  In Section 4.5, we visualize how a subclass is progressively pushed away from its superclass by an individual neuron during training.
>
> 3. If, as the reviewer suggest, randomness had a critical impact on the measure c_1 of a model, we would not observe a consistent correlation between c_1 and generalization performance. For example, models with poor generalization performance would sometimes achieve a higher c_1 value than models with good generalization performance because they were lucky to contain a neuron with the desired pattern. Thus, the max operation does not generate the problems that make the reviewer feel uncomfortable. To make this important observation clearer, we’ve dedicated a new Results Section to discuss this surprising observation (cfr. Section 4.2).
> Let us further motivate the introduction of the ‘max’ (or top-k mean) operation in metric c_1. It is not reasonable to expect that all neurons of a network differentiate all sub-classes, i.e. that each individual neuron defines a tight cluster for each of the sub-classes. Instead, it is more reasonable to assume that the neurons get the ability to specialize. Hence, when assessing the capability of the network to group a given sub-class C_i into a tight cluster, it is relevant to focus on the (k) neuron(s) that best discriminate(s) this particular sub-class from the others.
>
> 4.  We present 4 metrics, and *two of them actually do not require the intraclass clustering assignments* (c_3 and c_4). These two metrics c_3 and c_4 are the only ones to be considered on the datasets without available intraclass labels in our experiments. On datasets offering intra-class labelling, such as 20-class CIFAR-100, the four metrics are considered, and we indeed use the original label as intra-class assignment.
>
> 5. Same as above. Section 2.3 ‘Measures based on variance’, actually already implements what you are suggesting. We hope the clarification of this misunderstanding will lead to a positive revision of the reviewer’s rating.

---

### Author Response · Authors · 2020-11-19
**General message to the reviewers: thanks + summary of the new experimental results**

We thank the reviewers for their constructive comments. In particular, we very much appreciated the interest expressed by all reviewers towards the ideas introduced by our paper. The numerous questions raised to get a deeper understanding of the mechanisms underlying the observed correlation reveal by themselves the value of our message to the community.

It is beyond the scope of the paper to provide a solid and definitive explanation of the phenomenon underlying the observed correlation. However, as rightly pointed out by the reviewers, several experiments could deepen our understanding and strengthen the link between our measures and intraclass clustering. This criticism was expressed by 3 reviewers, and we took it very seriously during this rebuttal. We’ve thus performed numerous additional experiments and added them to the paper. The renewed result section contains the following six sections:

1)	(**already in initial submission**) Evaluating the measures' relationships with generalization.
2)	(**new**) Influence of k on the Kendall coefficients of neuron-level measures.
These results confer new insights around the sensitivity of the results to parameter k and the local nature of intraclass clustering. In particular, they suggest that each intraclass cluster is identified by a small sub part of the network, implying a form of specialization.
3)	(**new**) Evolution of the measures across layers.
These results suggest that intraclass clustering also occurs in the deepest representations of neural networks, and not merely in the first layers, which are commonly assumed to capture generic or class-independent features.
4)	(**already in initial submission**) Evolution of the measures over the course of training.
5)	(**new**) Visualization of subclass extraction in individual neurons.
These results show how a subclass is progressively differentiated from its superclass by an individual neuron during training, providing further evidence of the occurrence of intraclass clustering during training.
6)	(**new**) Discussion.
We restate how our different results support our work’s hypotheses.

As a whole, these new experiments add much value to our paper, and we thank the reviewers for highlighting their relevance. We also put a lot of effort to integrate the comments that were specific to each reviewer. These are detailed in our reviewer-specific replies.

---

### Decision · Program_Chairs · 2021-01-07
**Final Decision**

**Decision:**

Accept (Poster)

**Comment:**

The paper proposes intra-class clustering as an indicator of generalization performance and validates this by extensive empirical evaluation. All reviewers have found this connection highly interesting. The author response has also duly addressed most of the reviewers' concerns. Given the importance of studying generalization performance of overparameterized deep models, the paper will potentially generate interesting discussion at the conference.